# Sexual Dimorphism in the Limb Bones of Asiatic Toad (*Bufo gargarizans*) in Relation to Sexual Selection

**DOI:** 10.3390/ani13162638

**Published:** 2023-08-16

**Authors:** Chengzhi Yan, Hui Ma, Yuejun Yang, Zhiping Mi

**Affiliations:** Key Laboratory of Southwest China Wildlife Resources Conservation (Ministry of Education), China West Normal University, Nanchong 637009, China; ychzhi@cwnu.edu.cn (C.Y.); mahui102@cwnu.edu.cn (H.M.); 212021071300001@stu.cwnu.edu.cn (Y.Y.)

**Keywords:** body size, limb bones, sexual dimorphism, sexual selection, *Bufo gargarizans*

## Abstract

**Simple Summary:**

In this study, the sexual dimorphism of the limb bones of the Asiatic toad was investigated. Despite male toads having smaller body lengths than females, they possess longer upper arms, forearms, thighs, calves, and feet. Additionally, the humerus and radioulna bones of males are heavier compared to females, and the protrusions on the humerus and femur of males are longer and higher than females. However, female toads have significantly longer hands than males. These sexual differences in limb morphology are believed to be adaptations to specific reproductive behaviors. We propose that the sexual dimorphism in the limb morphology of the Asiatic toad may be attributed to amplexus and male–male competition during the long process of adaptive evolution.

**Abstract:**

Sexual dimorphism is often considered to be the result of differences in the intensity of sexual selection between sexes. From this point of view, the sexual dimorphism of the limb bones of the *Bufo gargarizans* in southwest China was studied. Results showed that the fore- and hindlimb skeletons of this species were sexually dimorphic in anatomy. The humerus, radioulna, and total lengths of the forelimb skeleton of males were substantially longer than those of females, but the hand length of males was smaller than that of females. Several other features of males, such as deltoid and medial crest areas and humerus and radioulnar weights, were also significantly larger than those of females. The femoris, tibiofibula, talus–calcaneus, and foot lengths; total hindlimb skeleton length; and femoral upper crest areas of males were significantly greater than those of females. However, no significant intersexual difference in femoris and tibiofibular weights was observed. These findings suggested that robust forelimb bones and long hindlimb bones could contribute to the mating success of males; if so, sexual selection promotes the evolution of sexual size and shape dimorphism in the limb bones of the *B. gargarizans*.

## 1. Introduction

Sexual size dimorphism (SSD), which refers to the difference in body size or mass between the sexes, is common in anurans, and in 90% of species females have a longer body length (i.e., snout-to-vent length (SVL)) than males [1,2,3,4]. Besides SSD, another common occurrence in anurans is sexual shape dimorphism (SShD), which pertains to differences in body shape between males and females [5]. The main mechanisms responsible for the evolution of sexual dimorphism (SD), such as sexual size dimorphism (SSD) and sexual shape dimorphism (SShD), are sexual selection and natural selection [6,7]. Fundamentally, SD is explained as a result of fecundity selection, intrasexual competition, niche divergence, and intersexual differences in life-history traits, such as age at maturity, growth rate, and longevity [8]. Additionally, ecological causation may play a significant role in shaping the degree of sexual size dimorphism [9]. Moreover, there is a prevailing belief that sexually selected traits typically exhibit positive static allometries, reflecting variation in shape among individuals of the same species [10].

The SVL of females is larger than that of males, but males still surpass females in some body dimensions. For example, in some species of anurans, which have an amplexus phenomenon in the breeding period, the fore- and hindlimbs of males exhibit sexual dimorphism in morphology and contractile characteristics. Males have longer and heavier muscles, more muscle fibers, higher muscle forces, longer and heavier humeri and radioulnas, and larger areas for muscle attachment on the humerus than females [11,12,13,14,15,16]. Previous studies found that the relative or absolute weight of males’ forelimb muscles associated with amplexus and hindlimb muscles, which are related to the scramble competition, are larger than that of females; however, the weight of the forelimb muscles, which is unrelated to the clasping action, does not differ between sexes [12,14,15,16,17,18,19,20,21]. According to the sexual selection theory, such secondary sexual dimorphism in limbs is attributed to adaptation for amplexus, which is a behavior used in male–male competition [15,21,22]. However, bones are the attachment of muscles and the lever of movement, and knowledge on the differences in limb bone anatomy between the sexes is lacking [12,23].

The Asiatic toad (*Bufo gargarizans* Cantor, 1842) is widely distributed in China, Russia, and the Korean Peninsula [24]. The Asiatic toad is an explosive breeder, and reproduction takes place in December–January or January–February with a breeding period of 25–30 days in Nanchong City, China [25]. In addition, some information on the morphological anatomy, larval development, habitat selection, diet, mating behavior, age structure, brain size evolution, muscular metabolism, and secondary sexual dimorphism of the *B. gargarizans* has been reported in recent years [26,27,28,29,30,31,32,33]. This study aims to document the traits of sexual dimorphism in the size and shape of the limb bones of the species and explore implications of observed differences for the evolutionary ecology of the *B. gargarizans*.

## 2. Materials and Methods

### 2.1. Specimen Preparation and Measurement

Specimens were collected from Jintai Town, Nanchong City (30°93′ N, 106°12′ E, 319 m above sea level), Sichuan Province, China. A total of 37 toads, which included 20 males and 17 females, were captured on the evening of 2 April 2019. The collection of toads followed all the applicable instructions of the institutional research ethics committee of China West Normal University (code: 20190122). The toads were temporarily kept in an artificial tank (L × W × H = 0.6 m × 0.5 m × 0.8 m) before processing. All individuals were killed through the double-pithing method. The SVL was measured using an electronic caliper to the nearest 0.01 mm, and the body weight was recorded to the nearest 0.01 mg using an electronic balance. The fore- and hindlimbs from the right side of each specimen were dissected between 3 and 8 April 2019. After the skin was stripped off, the right fore- and hindlimbs of the *B. gargarizans* were completely taken down. The muscles attached to the bone were removed to the greatest extent using scissors and a scalpel. Hands (i.e., metacarpus and digiti manus) and feet (i.e., metatarsus and digiti pedis) were cut, and the remaining limbs were soaked in 4% NaOH at 50 °C for 5 min, collected, washed with running water, and brushed carefully with a toothbrush to remove the residual connective tissues and muscles on the bone. In accordance with the above method, the humerus, radioulna, femoris, tibiofibula, and talus–calcaneus specimens were prepared. The carpi, metacarpi, digiti manus, small tarsi, metatarsi, and digiti pedis specimens were not prepared because they are extremely small. The sum of the lengths of the 4th metacarpus and the 4th finger was used as the hand length, and the sum of the lengths of the 4th metatarsus and the 4th toe was used as foot length. The maximum length of each bone was measured to the nearest 0.01 mm, and the bone was placed into a piece of filter paper to dry the surface and weighed to the nearest 0.01 mg. Measurements were made using an electronic caliper and balance. The deltoid and medial crests of the humerus and femoral upper crest are important points of muscle attachment. Thus, their maximum lengths and widths were recorded, and the area of each crest was approximately estimated as the product of these two dimensions. In order to minimize human error in length and width measurements, a specific person was dedicated to measuring each part twice, and the average value was computed. The diagrams of bones examined as above are shown in Figure 1. In addition, the longest fourth toe of the left hindlimb was cut off and stored in a 10% formalin solution for skeletochronological analysis. The skeletochronological procedure was performed according to our previous work [34].

### 2.2. Statistical Analysis

First, data were subjected to the Kolmogorov–Smirnov test to determine the normal distribution. Differences in SVL and age between males and females were tested using a one-way ANOVA, respectively. The correlation between age and SVL was analyzed using the Spearman method. General linear models were used with age as a covariate to see whether differences in SVL between sexes remained after removing the effects of age. The bone length, crest areas, bone weight, and hand and foot lengths were regressed on the SVL, and slopes (β) between sexes were compared using regression analysis to test for homogeneity. Then, the significance of differences in these indices between sexes was tested through the multiple dependent variables analyses of covariance (MANCOVA) using the SVL and age as covariates. Statistical analyses were performed using the SPSS 23.0, and the significance level was set to 0.05. Data were presented as mean ± SD.

## 3. Results

### 3.1. Sexual Differences in Age and Snout-to-Vent Length

The age ranged from 2 to 5 yrs in both males and females. The age structure is shown in Figure 2. Individuals of 3 and 4 years old were the most numerous in both sexes. There was no significant difference between the average ages of both sexes (male: *n* = 20, mean = 3.20 ± 0.89; female *n* = 17, mean = 3.53 ± 1.01; ANOVA, *F*_1,36_ = 1.11, *p* = 0.299).

The SVL ranged from 95.95 mm to 110.96 mm (*n* = 17, mean = 101.17 ± 4.90) in females and from 86.18 mm to 111.15 mm (*n* = 20, mean = 96.76 ± 6.35) in males. The average SVL of females was 1.05 times longer than that of males (ANOVA, *F*_1,36_ = 5.45, *p* = 0.025). The SVL was greatly correlated with age in both sexes (Spearman’s correlation coefficients, male: *r_s_* = 0.94, *n =* 20, *p* < 0.001; female: *r_s_* = 0.91, *n =* 17, *p* < 0.001). In order to remove the effects of age, we ran a general linear model (GLM) with the SVL as the dependent variable, sex as the independent variable, and age as a covariate. The results showed that there was still a significant difference in the SVL between the sexes, with females larger than males (ANCOVA, *F*_1,35_ = 8.48, *p* = 0.006).

### 3.2. Comparative Analysis of the Forelimb Skeleton

The forelimb skeleton comprises the humerus, radioulna, carpus, metacarpus, and digitorum manus in both sexes (Figure 3a–c). The total length of the forelimb skeleton of males was longer than that of females. Notably, the proximal half of the humerus exhibited a prominent deltoid crest on its ventral surface, while the distal half featured a medial crest along its inner margin, with the deltoid crest and medial crest of males being more pronounced than those of females (Figure 3a). The radioulna represented a fusion of the radius and ulna (Figure 3b). The humerus and radioulna of males were longer and heavier than those of females. The carpus was composed of six diminutive bones, arranged in two rows: three proximal and three distal. Furthermore, there were five metacarpal bones, with the first metacarpal being particularly small in size. Regarding the digits, the first finger degenerated, while the second and third fingers possessed two phalanges each; the fourth and fifth fingers had three phalanges, respectively (Figure 3c). The hand length was shorter in males than in females.

The linear regression statistics of the humerus, radioulna, and hand lengths; deltoid and medial crest areas; and humerus and radioulnar weights on the SVL for males and females and results of comparing slopes (*β*) between the sexes are shown in Table 1. Except the humerus length and medial crest area of females, the linear regression of other skeletal elements of the forelimbs on the SVL for both sexes was highly significant (*p* < 0.05), and the slopes between males and females were homogeneous (*p* > 0.05). In every case, the differences between the means of males and females were highly significant, as shown by MANCOVA (Table 2). When the influence of the SVL and age were controlled, the humerus and radioulnar lengths; deltoid and medial crest areas; and humerus and radioulnar weights of males significantly exceeded 1.10–3.29 times those of females, but the hand length of males was 90.7% that of females (Table 2). The total length of the forelimb skeleton regressed significantly on SVL for each sex (Figure 4; males: *n* = 20, *R*^2^ = 0.657, *β* = 0.810, *F* = 34.44, *p* < 0.001; females: *n* = 17, *R*^2^ = 0.345, *β* = 0.587, *F* = 7.90, *p* = 0.013), and the slopes were homogeneous (*t* = −0.200, *p* = 0.842). Independent of body size and age, the total length of the forelimb skeleton of males exceeded 1.04 times that of females (Table 2).

### 3.3. Comparative Analysis of the Hindlimb Skeleton

The hindlimb skeleton encompassed femoris, tibiofibulas, tarsi, metatarsi, and digitorum pedis (Figure 3d–g). The total length of the hindlimb skeleton was longer in males than in females. The ventral surface of the proximal half of the femoris exhibited a prominent crest referred to as the femoral upper crest, with the femoral upper crest of males being more pronounced than females (Figure 3d). The tibiofibula was a fusion of the tibia and fibula (Figure 3e). The tarsi consisted of five bones, with the proximal calcaneus and talus merging at both ends (known as the talus–calcaneus) (Figure 3f). The lengths of the femoris, tibiofibular, and talus–calcaneus were all longer in males than in females, but their weights did not differ between sexes. In contrast, the three distal tarsus bones were extremely diminutive in size. The metatarsi consisted of five elements, with the fourth metatarsal being the longest. The digitorum pedis, in turn, demonstrated distinct phalangeal compositions across the digits: the first and second digits possessed two phalanges; the third and fifth digits comprised three phalanges; and the fourth digit exhibited four phalanges (Figure 3g). The foot length of males was longer than that of females.

The linear regression of eight skeletal features (femoris, tibiofibula, talus–calcaneus, and foot lengths; femoris, tibiofibula, and talus–calcaneus weights; and femoral upper crest area) on the SVL of males and females was highly significant (*p* < 0.05), and the slopes were homogeneous for all comparisons between sexes (*p* > 0.05). Details are summarized in Table 3. MANCOVA showed that the five skeletal features (i.e., femoris, tibiofibula, talus–calcaneus, and foot lengths and the femoral upper crest area) of males significantly exceeded 1.02–1.13 times those of females. Especially, although the talus–calcaneus weight of males statistically was smaller than that of females, in fact, the difference in their absolute weight was very small (Table 4). However, the femoris and tibiofibula weights did not differ statistically between sexes (Table 4). The total length of the hindlimb skeleton regressed significantly on the SVL for both sexes (Figure 5; males: *n* = 20, *R*^2^ = 0.740, *β* = 0.860, *F* = 51.14, *p* < 0.001; females: *n* = 17, *R*^2^ = 0.440, *β* = 0.663, *F* = 11.76, *p* = 0.004) with homogeneous slopes (*t* = 0.405, *p* = 0.688). Independent of body size and age, the total hindlimb skeleton length of males exceeded 1.06 times that of females (Table 4).

## 4. Discussion

Like in most anurans, the SVL of the female *B. gargarizans* exceeds that of the male *B. gargarizans*, and this finding is in line with those observed in previous studies on this species [18,27,34]. Previous studies have revealed that sexual size dimorphism (SSD) may be related to age structure, age at sexual maturity, growth rate, and fecundity [2,35,36,37,38,39,40]. For the *B. gargarizans*, males reach maturity one year earlier than females [34]. According to resource allocation theory, this early sexual maturity in males may result in a higher allocation of resources towards reproductive traits, such as an increased sperm production or elaborate courtship behaviors, leaving fewer resources available for growth and body size [41,42,43]. This theory assumes that individuals have similar resource budgets, but it is important to acknowledge that there may be genetic variation in resource acquisition ability and positive correlations among fitness traits [44]. If males of high quality possess larger resource budgets, they can allocate more resources towards both overall size and reproductive characteristics. Moreover, correlations between survival and genetic quality or fitness can be established across all age groups, with early- and late-age survival and fecundity increasing in tandem with male quality [45]. Thus, it seems that the smaller body size of males, in comparison to females, cannot be solely attributed to males maturing earlier than females.

Given that females reach sexual maturity later than males, it is implausible to assume that females invest more in reproduction than males [42]. Rather, the differences in investment between the sexes can be attributed to the distinct strategies they employ to maximize reproductive success, such as gametogenesis and parental care [46,47]. The female *B. gargarizans* has longer longevity than the male *B. gargarizans* [34]; Old-aged females may attain greater body length because amphibians have the characteristics of lifelong growth [1], but, in this study, although the average age of females was older than that of males, there was no statistically significant difference, possibly due to the small sample size. In addition, a significantly positive correlation is observed between ovarian weight and SVL during the breeding season (our unpublished data), and larger female *B. gargarizans* specimens can lay more eggs [27]. The fecundity selection theory states that female-biased SSD is the result of fecundity selection, which is based on the positive correlation between female fecundity and body size [37,39]. Notably, differences in fecundity selection and longevity may be two important factors that lead to female-biased SSD in the *B. gargarizans*. However, knowledge on the differences in growth rate, ecological causation, and other unknown factors between the sexes of the *B. gargarizans* and their effect on the SSD of the *B. gargarizans* is limited and should be further studied in the future.

During the breeding season of the *B. gargarizans*, the male grasps the female’s axillary using his forelimbs. Mi revealed that the weight of forelimb muscles in the male *B. gargarizans* is larger than that in the female *B. gargarizans* [15]. In the present study, the humerus and radioulnar lengths; deltoid and medial crest areas on the humerus; humerus and radioulnar weights; and total forelimb skeleton length of the male *B. gargarizans* are larger than those of the female *B. gargarizans*. The long bones and large protrusions can provide suitable attachment for large muscles. A heavy forelimb bone can withstand substantial muscular pull produced by heavy muscles [13,48]. Thus, the structure and function of the bones and muscles of the forelimb of the *B. gargarizans* are compatible, and this finding is consistent with those of Lee. In addition, the longer and heavier humerus and radioulna of the male *B. gargarizans* can afford a secure grip of the female *B. gargarizans* [12]. Furthermore, this characteristic is expected to allow them to mate with larger females and consequently enhance their fertility [8]. The elongated forelimb in the male *B. gargarizans* may also enhance their dispersal ability, allowing them to search for potential mates across wider areas. This increased mobility could significantly improve their chances of finding a suitable partner for reproduction [49]. However, the male *B. gargarizans* has shorter hands than the female *B. gargarizans*, and this phenomenon is also found in the *Bufotes viridis* [23] and may be because the length of the male’s hand has little effect on amplexus success. The male *B. gargarizans* clasps the female’s axillary with the wrist and the back of the medial three fingers (i.e., thumb and second and third fingers), which have nuptial pads, instead of embracing the female with his fingers [50]. The female *B. gargarizans* carries the male *B. gargarizans* on her back during amplexus and has large hands for body support and easy movement. Therefore, in the *B. gargarizans*, selective forces seem to have favored the male humerus and radioulnar and the female metacarpal and phalangeal bones (hands) rather than entire forelimb bones. We posit that sexual selection associated with amplexus behavior is the driver of sexual size and shape dimorphism in the forelimb skeleton of the *B. gargarizans*.

In the present study, the femoris, tibiofibula, talus–calcaneus, and foot lengths; femoral upper crest area; and total hindlimb skeleton length of the male *B. gargarizans* are larger than those of females. The long femoris, tibiofibula, talus–calcaneus, and foot can provide an effective lever arm for the movement of several hindlimb joints. A long foot may also afford a speed advantage [51]. Males with long hind legs and feet have the advantages of kicking, jumping, and swimming. These physical attributes may grant them the potential to elude predators swiftly, quickly find a mate, and effectively drive away competitors from the back of females [17,52]. Furthermore, Mi found that the hindlimb muscles of the male *B. gargarizans* are heavier than those of the female *B. gargarizans* [19]. Thus, the long hindlimb skeleton and large femoral upper crest provide an adaptive attachment surface for the massive hindlimb muscles in males. These findings indicate that the long hindlimbs of males may provide selective advantages for reproductive success. The high male–male competition in the *B. gargarizans* can produce long hindlimbs via sexual selection [27].

While we have uncovered sexual dimorphism in the limb bones of the *B. gargarizans*, it is important to note that our findings have certain limitations. Specifically, we did not take into account the influence of altitudes and habitats on the sexual dimorphism in the *B. gargarizans*. As a result, it is advisable to interpret our results with caution, considering the limited sample size.

## 5. Conclusions

The total length of the forelimb bones of the male *B. gargarizans* is longer than that of the female *B. gargarizans* because of the longer humerus and radioulna and shorter hand length of males compared to those of females. In males, large deltoid and medial crest areas on the humerus provide attachment for heavy muscles, and the heavy humerus and radioulna can bear the high tension produced by large muscles. We hypothesize that a robust forelimb may be the result of adaptation to amplexus during the breeding season. The total male hindlimb skeleton and its components (i.e., femoris, tibiofibula, talus–calcaneus, and foot) are longer than those of females, thereby providing males an advantage in finding a mate and resisting displacement. Sexual selection pressure has possibly favored the bones and muscles associated with reproductive success. Perhaps sexual selection can be used to explain the sexual size and shape dimorphism of the limb bones of the *B. gargarizans*. However, the influence of other factors, such as ecological factors, on the sexually dimorphic limb bones of the *B. gargarizans* cannot be dismissed.

## Figures and Tables

**Figure 1 animals-13-02638-f001:**
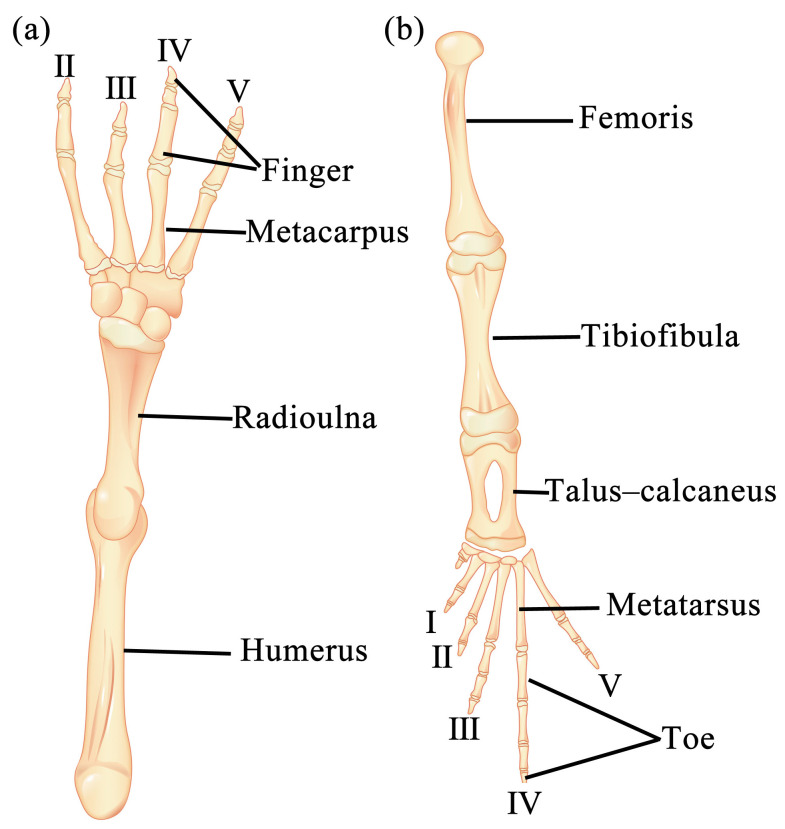
Diagrams showing the anatomy of forelimb (**a**) and hindlimb bones (**b**) of *B. gargarizans* and locations where the samples were taken.

**Figure 2 animals-13-02638-f002:**
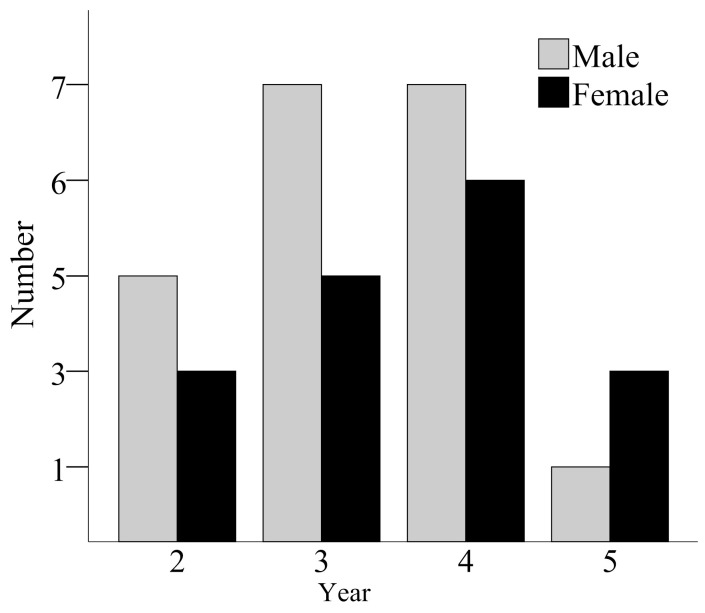
Age structures of males and females from a population of *B. gargarizans*. The numbers of males at ages 2, 3, 4, and 5 are 5, 7, 7, and 1, respectively, and the numbers of females at ages 2, 3, 4, and 5 are 3, 5, 6, and 3, respectively.

**Figure 3 animals-13-02638-f003:**
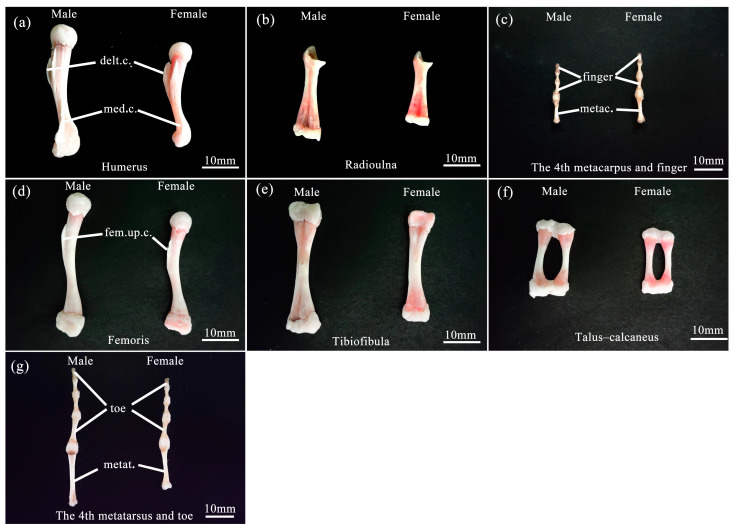
Comparison of limb bones features of male and female *B. gargarizans.* (**a**–**c**) represent the humerus, radioulna, 4th metacarpus, and 4th finger in the forelimb bones, respectively. (**d**–**g**) represent the femoris, tibiofibula, talus–calcaneus, 4th metatarsus, and 4th toe in the hindlimb bones, respectively. Abbreviation: delt.c., deltoid crest; med.c., medial crest; metac., metacarpus; fem.up.c., femoral upper crest; metat., metatarsus. Bars = 10 mm.

**Figure 4 animals-13-02638-f004:**
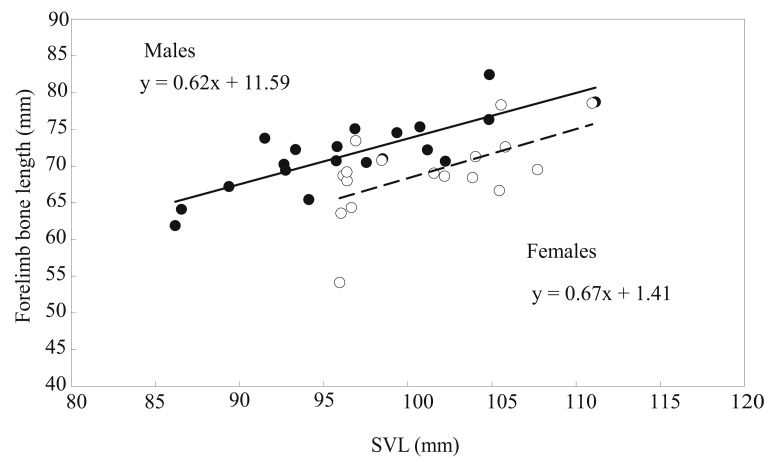
Linear regression of the total length of forelimb bones on the SVL of male (closed circles, solid line) and female (open circles, broken line) *B. gargarizans*.

**Figure 5 animals-13-02638-f005:**
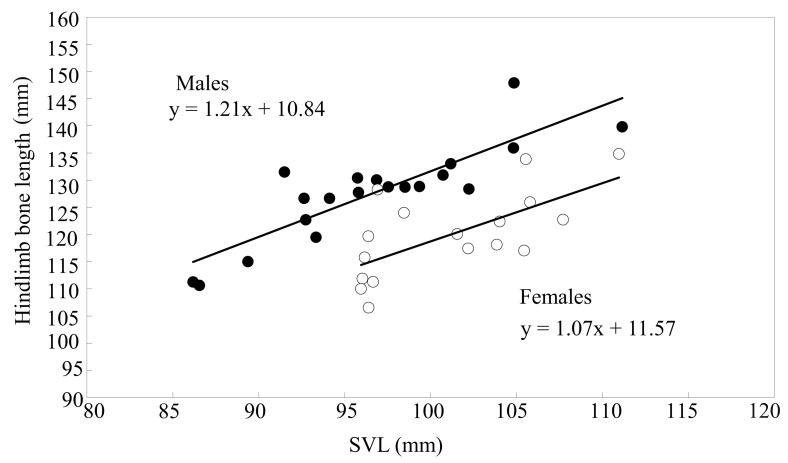
Linear regression of the total length of hindlimb bones on the SVL of male (closed circles, solid line) and female (open circles, broken line) *B. gargarizans*.

**Table 1 animals-13-02638-t001:** Linear regression statistics of forelimb bone size on SVL and results of slope homogeneity comparison of male and female *B. gargarizans.* M, male (*n* = 20); F, female (*n* = 17).

		Regression Statistics					Coefficient M *β* ≠ F *β*	
Variable	Sex	Equation	*R* ^2^	*β*	*F*	*p*	*t*	*p*
Humerus length	M	y = 0.33x + 0.56	0.671	0.819	36.66	0.000		
	F	y = 0.32x − 3.00	0.230	0.480	4.49	0.051		
Humerus weight	M	y = 22.09x − 1443.11	0.724	0.851	47.32	0.000	1.93	0.063
	F	y = 11.78x − 579.20	0.348	0.590	8.01	0.013		
Deltoid crest area	M	y = 1.32x − 93.67	0.686	0.828	39.38	0.000	1.58	0.123
	F	y = 0.77x − 48.75	0.362	0.601	8.50	0.011		
Medial crest area	M	y = 1.39x − 117.13	0.654	0.809	34.02	0.000		
	F	y = 17.02 − 0.12x	0.029	−0.172	0.46	0.510		
Radioulnar length	M	y = 0.17x + 2.84	0.708	0.841	43.61	0.000	0.89	0.379
	F	y = 0.13x + 4.70	0.375	0.612	9.00	0.009		
Radioulnar weight	M	y = 8.88x − 599.33	0.641	0.800	32.10	0.000	0.96	0.343
	F	y = 6.33x − 412.99	0.379	0.615	9.14	0.009		
Hand length	M	y = 0.12x + 8.19	0.279	0.528	6.97	0.017	−1.06	0.296
	F	y = 0.22x − 0.30	0.309	0.556	6.72	0.020		

**Table 2 animals-13-02638-t002:** Means and standard deviations of forelimb bone size and results of MANCOVA comparing male and female *B. gargarizans*.

	Males (*n* = 20)		Females (*n* = 17)		MANCOVA	
Variable	Mean ± SD	Range	Mean ± SD	Range	*F*	*p*
Humerus length (mm)	31.97 ± 2.52	26.65–37.60	29.09 ± 3.24	17.92–32.22	23.27	0.000
Humerus weight (mg)	694.00 ± 164.68	400.00–1040.00	612.35 ± 97.82	480.00–800.00	30.31	0.000
Deltoid crest area (mm^2^)	33.89 ± 10.10	6.73–49.14	29.19 ± 6.28	18.72–40.72	22.06	0.000
Medial crest area (mm^2^)	17.00 ± 10.88	0.72–47.33	5.16 ± 3.34	1.25–15.62	43.46	0.000
Radioulnar length (mm)	19.55 ± 1.30	17.29–22.42	17.77 ± 1.03	16.31–20.25	71.28	0.000
Radioulnar weight (mg)	260.00 ± 70.41	140.00–420.00	227.65 ± 50.44	170.00–350.00	23.16	0.000
Hand length (mm)	20.20 ± 1.49	16.37–22.43	22.27 ± 1.97	19.67–26.88	5.03	0.032
Total forelimb skeleton length (mm)	71.73 ± 4.87	61.86–82.45	69.13 ± 5.59	54.12–78.57	13.45	0.001

**Table 3 animals-13-02638-t003:** Linear regression statistics of hindlimb bone size on SVL and results of slope homogeneity comparison of male and female *B. gargarizans*. M, male (*n* = 20); F, female (*n* = 17).

		Regression Statistics					Coefficient M *β* ≠ F *β*	
Variable	Sex	Equation	*R* ^2^	*β*	*F*	*p*	*t*	*p*
Femoris length	M	y = 0.35x + 1.48	0.486	0.697	17.01	0.001	0.93	0.360
	F	y = 0.23x + 11.19	0.336	0.580	7.60	0.015		
Femoris weight	M	y = 16.93x − 995.80	0.619	0.787	29.21	0.000	0.84	0.409
	F	y = 12.11x − 564.36	0.280	0.529	5.83	0.029		
Femoral upper crest area	M	y = 0.65x − 43.95	0.313	0.560	8.21	0.010	0.42	0.676
	F	y = 0.51x − 34.65	0.315	0.562	6.91	0.019		
Tibiofibular length	M	y = 0.28x + 5.21	0.708	0.842	43.67	0.000	0.40	0.695
	F	y = 0.25x + 5.50	0.483	0.695	14.03	0.002		
Tibiofibular weight	M	y = 16.86x − 1020.70	0.658	0.811	34.63	0.000	1.04	0.308
	F	y = 11.42x − 522.29	0.295	0.543	6.28	0.024		
Talus–calcaneus length	M	y = 0.16x + 3.59	0.573	0.757	24.11	0.000	0.23	0.818
	F	y = 0.14x + 3.56	0.371	0.609	8.83	0.009		
Talus–calcaneus weight	M	y = 9.74x − 607.33	0.599	0.774	26.92	0.000	0.58	0.563
	F	y = 7.84x − 457.04	0.365	0.604	8.63	0.010		
Foot length	M	y = 0.42x + 0.57	0.587	0.766	25.60	0.000	−0.14	0.887
	F	y = 0.45x − 8.67	0.283	0.532	5.92	0.028		

**Table 4 animals-13-02638-t004:** Means and standard deviations of hindlimb bone size and results of MANCOVA comparing male and female *B. gargarizans*.

	Males (*n* = 20)		Females (*n* = 17)		MANCOVA	
Variable	Mean ± SD	Range	Mean ± SD	Range	*F*	*p*
Femoris length (mm)	35.13 ± 3.17	26.17–41.78	34.39 ± 1.94	31.61–38.22	7.50	0.010
Femoris weight (mg)	642.00 ± 136.56	380.00–1010.00	661.18 ± 112.19	510.00–900.00	2.80	0.104
Femoral upper crest area (mm^2^)	18.77 ± 7.35	6.72–35.07	16.86 ± 4.44	11.16–27.77	6.16	0.018
Tibiofibular length (mm)	32.73 ± 2.14	28.69–37.72	31.13 ± 1.79	28.07–34.42	36.72	0.000
Tibiofibular weight (mg)	610.50 ± 131.89	370.00–940.00	632.94 ± 103.00	500.00–880.00	3.42	0.073
Talus–calcaneus length (mm)	18.70 ± 1.31	16.25–21.22	18.03 ± 1.15	16.76–20.19	15.95	0.000
Talus–calcaneus weight (mg)	335.00 ± 79.84	190.00–520.00	335.89 ± 63.55	240.00–480.00	6.09	0.019
Foot length (mm)	41.16 ± 3.47	33.95–47.18	36.44 ± 4.11	25.62–43.75	38.97	0.000
Total hindlimb skeleton length (mm)	127.72 ± 8.91	110.62–147.90	119.99 ± 7.92	106.54–134.82	42.99	0.000

## Data Availability

The authors confirm that the data presented in this study are available in Appendix A.

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
