# Peer review of "Sexual Dimorphism in the Limb Bones of Asiatic Toad (Bufo gargarizans) in Relation to Sexual Selection"

_animals, 2023, doi:10.3390/ani13162638_

Round 1

Reviewer 1 Report

General Comments

This paper establishes patterns of sexual dimorphism in boney characters of Bufo gargarizans. Such patterns suggest that these morphological characters may be under sexual selection. However, the authors need to be more careful with over interpreting these results. Future work is necessary to establish that sexual selection is the mechanism responsible for the observed patterns. Ecological alternatives to sexual selection need to be introduced in the paper and discussed.

The authors may want to consider using a MANCOVA to analyze these data. Multiple dependent variables with sex and body size (SVL) as predictors is a data set that seems perfect for MANCOVA. The results produced may be less cumbersome than running multiple ANCOVAs and t-tests—not to mention the problem with inflating alpha.

I also recommend that the authors provide a magnitude for differences when sharing results. Rather than writing “Character A was larger in males than females.” I recommend “Character A was X times or X% larger in males than females.” This provides the reader with an idea of how different the patterns of dimorphism were across characters in this study.

Below are some specific edits for the authors to consider and some papers that I think would improve the introduction and discussion.

Specific Comments

Line 17: “Sexual dimorphism often…..”

Line 20: “…..species were sexually dimorphic…”

Line 22: “Several other features,……”

Lines 28-29: “…..mating success of males. If so, ……..”

Lines 34-35: “….anurans, and in 90% of species females have a longer body…..” 

Lines 45-47: “According to sexual selection theory………for amplexus which is a behavior used in male-male…”

Line 63: delete “was”

Lines 65: capitalize name of university

Line 75: delete “it”

Line 102: replace “higher” with “longer”

Line 109: I think it is more informative to report that slopes were equal rather than referring to beta values. In the introduction and discussion, it would be nice if you provided context for when slopes may differ between the sexes. Here is a nice review paper that covers the topic:

Bonduriansky, Russell. "Sexual selection and allometry: a critical reappraisal of the evidence and ideas." Evolution 61.4 (2007): 838-849.

Lines 160-164: I found this section difficult to follow. I think the authors are suggesting that investment in reproductive characters may limit body size due to allocation tradeoffs. However this assumes that individuals have similar resource budgets. If high quality males have larger resource budgets, they can invest more in both overall size and reproductive characters.

Reznick, David, Leonard Nunney, and Alan Tessier. "Big houses, big cars, superfleas and the costs of reproduction." Trends in ecology & evolution 15.10 (2000): 421-425.

Kokko, Hanna. "Good genes, old age and life-history trade-offs." Evolutionary Ecology 12 (1998): 739-750.

I also do not follow the argument that females invest more in reproduction than males. It is often just that the kinds of characters the sexes invest in often differs because of the strategies they use to maximize reproductive success. 

Wedell, Nina, Matthew JG Gage, and Geoffrey A. Parker. "Sperm competition, male prudence and sperm-limited females." Trends in ecology & evolution 17.7 (2002): 313-320.

Kokko, Hanna, and Michael Jennions. "It takes two to tango." Trends in Ecology & Evolution 18.3 (2003): 103-104.

Lines 206-208: need a citation here that documents strong male-male competition in this species.

Line 207: “may provide”

Lines 214-215: need a qualifier here. May be true but need to do the work to test the hypothesis.

Line 218: delete “to”

Lines 218-220: These statements are too definitive given the evidence presented in the paper. Sure, sexual dimorphism is often due to sexual selection but not always. The following paper should be cited and used in the introduction and discussion.

Shine, Richard. "Ecological causes for the evolution of sexual dimorphism: a review of the evidence." The Quarterly review of biology 64.4 (1989): 419-461.

Literature cited: Please double check references. For example, there are several cases where journal titles are not capitalized.

Author Response

Dear reviewers, thank you for your careful review and constructive suggestions regarding our manuscript. We have revised the manuscript in accordance with the comments and marked all the amends on our revised manuscript. Our responses to the specific comments from reviewers are uploaded as an attachment.

Reviewer 2 Report

1.        The manuscript investigated sexual dimorphism of the limb bones of the Bufo gargarizans. The study noted that robust forelimb bones and long hindlimb bones in males suggest the adaptation to amplexus behavior during breeding and/or enhance the chances of finding a mate. As one of the widely distributed species, Bufo gargarizans may face different selection pressures in different altitudes and habitats, so it is recommended to consider more sample sizes to enrich the conclusion.

2.        The study aims to document the limb bones size and shape of the species and explore the implications of observed differences for the evolutionary ecology. Therefore, the comparison of limb bone shape between the sexes should be supplemented by more figure presentation and description.

3.        It is recommended to add anatomical drawings of bones and muscles after materials and methods to visually demonstrate the measured region.

4.        Previous studies have revealed that sexual size dimorphism (SSD) may be related to age structure, age at sexual maturity, growth rate and fecundity. What is the effect of toad age on muscle growth? Whether age causes changes in bone and muscle weight, please provide bone age data on the basis of existing bone specimens for size and shape analysis and discussion.

5.        The study describes differences in the size and shape of the bones and muscles of the forelimbs and hindlimbs of this species, but does not cover a comparison of all the muscle and bone features of the limbs (carpi, metacarpi, digiti manus, small tarsi, metatarsi and digiti pedis), and the discussion section is slightly insufficient.

  • No obvious language errors were found in the full text, and a small amount of English  editing is recommended.

Author Response

(The authors gave the same response as above.)

Round 2

Reviewer 2 Report

The manuscript uses statistical and anatomical methods to explore differences in head length and limb bones between the sexes in B. gargarizans. The study pointed out that there were significant sex differences in both svl and fore-hind limb bones (svl correction), and the possible factors for the differences were discussed. After revision, the method, conclusion and discussion of the manuscript have been greatly improved, but there are still some small problems that can be improved. To further improve this manuscript, I have some notes in manuscript and the following suggestions:

(1)Page1This study mainly explores the relationship between morphology and sexual selection. The ‘Introduction’ section proposes to supplement the literature on the causes of morphological dimorphism (sexual selection, fertility selection, etc.) and whether SVL dimorphism is related to sexual selection. Sexual selection is related to ecology and allometric growth, but these two are not covered below, and it is recommended to reduce in length or mentioned in a discussion.

(2)Page 3

It is recommended to put the results of age comparison in the first paragraph and the results of svl differences in the second paragraph, because the differences in svl are compared on the basis of age.

(3)Page4 and Page7

Sections 3.2 and 3.3 are titled ‘Comparative analysis of the skeleton’, however, in the first paragraph of each section, there is no comparison of gender differences in limb morphology.

(4)Page5

In Figure 3, is there calcified cartilage at both ends of the limbs? Are there significant sexual differences in morphology? It is recommended to increase the comparison of cartilage parts.

(5)Please add scale in Figure 3.

(6)In ‘Discussion’, pleased add the functional or adaptive significance of differences in gender limb morphology in the discussion section.

Note: The Latin names of species in references are italicized.

English language fine.

Author Response

Dear reviewers, thank you for your careful review and constructive suggestions regarding our manuscript. The comments are all valuable and very helpful for revising and improving our paper, as well as the important guiding significance to our researches. We have revised the manuscript in accordance with the comments and marked all new amends in blue on our revised manuscript. Our responses to the specific comments from reviewers are uploaded as an attachment
